

# A robust automated technique for operational calibration of ceilometers using the integrated backscatter from totally attenuating liquid clouds

Emma Hopkin[1], Anthony J. Illingworth[1], Cristina Charlton-Perez[2], Chris D. Westbrook[1], Sue
Ballard[2,3]

[1]Department of Meteorology, University of Reading, Reading, UK
[2]Met Office, MetOffice@Reading, Reading, UK
[3]Deceased, 12th July 2018

*Correspondence to:* Emma Hopkin (EH@westgate.slough.sch.uk); Dr. C. Charlton-Perez (c.charlton-
perez@metoffice.gov.uk)

**Abstract.** A simple and robust method for calibrating ceilometers has been tested in an operational environment
demonstrating that the calibrations are stable to better than ± 5% over a period of a year. The method relies on
using the integrated backscatter (B) from liquid clouds that totally extinguish the ceilometer signal; B is
inversely proportional to the lidar ratio (S) of the backscatter to the extinction for cloud droplets. The calibration
technique involves scaling the observed backscatter so that B matches the predicted value for S of $18.8 \pm 0.8$ sr
for cloud droplets, at ceilometer wavelengths. For accurate calibration, care must be taken to exclude any
profiles having targets with different values of S, such as drizzle drops and aerosol particles, profiles that do not
totally extinguish the ceilometer signal, profiles with low cloud bases that saturate the receiver, and any profiles
where the window transmission or the lidar pulse energy is low. A range dependent multiple scattering
correction that depends on the ceilometer optics should be applied to the profile.  A simple correction for water
vapour attenuation for ceilometers operating at around 910 nm wavelength is applied to the signal using the
vapour profiles from a forecast analysis.  For a generic ceilometer in the UK the 90-day running mean of the
calibration coefficient over a period of 20 months is constant to within 3% with no detectable annual cycle, thus
confirming the validity of the humidity and multiple scattering correction.  For Gibraltar, where cloud cover is
less prevalent than in the UK, the 90-day running mean calibration coefficient was constant to within 4%. The
more sensitive ceilometer model operating at 1064 nm is unaffected by water vapour attenuation but is more
prone to saturation in liquid clouds. We show that reliable calibration is still possible, provided the clouds used
are above a certain altitude. The threshold is instrument dependent but is typically around 2 km. We also
identify a characteristic signature of saturation, and remove any profiles with this signature. Despite the more
restricted sample of cloud profiles, a robust calibration is readily achieved, and, in the UK, the running mean 90-
day calibration coefficients varied by about 4% over a period of one year.  The consistency of profiles observed
by nine pairs of co-located ceilometers in the UK Met Office network operating at around 910 nm and 1064 nm
provided independent validation of the calibration technique. EUMETNET is currently networking 700
European ceilometers so they can provide ceilometer profiles in near real time to European weather forecast
centres and has adopted the cloud calibration technique described in this paper for ceilometers with a
wavelength of around 910 nm. (439 words)



## 1 Introduction

Ceilometers are simple, relatively inexpensive vertically pointing lidars that typically operate at wavelengths of
905-910 nm or 1064 nm. They can be left unattended for long periods and, as the name suggests, have mainly
been used for detecting cloud base height at airports where they are valuable for air safety issues. Recent studies
have shown that, in addition to detecting the large backscattered return signal from cloud base, they can also
provide vertical profiles of backscatter from both clouds and aerosols every 5-30 seconds with a range
resolution as low as 10 m. Ceilometer profiles have been used  in many research contexts; some examples are
for Cloudnet scheme for validation of the representation of clouds in operational numerical weather prediction
(NWP) forecast models (Illingworth et al., 2015), for aerosol profiling (Markowicz et al., 2008; Madonna et al.,
2015), fog observations (Dupont et al., 2012) and the retrieval of mixing height levels (Münkel et al., 2007).

Many operational weather forecasting models now represent both clouds and aerosols by prognostic variables.
Remote sensing observations are needed to show that these models are providing unbiased estimates of aerosol
and cloud properties, and ultimately for data assimilation into such models to improve forecasts of hazardous
weather such as pollution episodes and severe convective storms producing flash floods. The European Ground-
Based Observations of Essential Variables for Climate and Operational Meteorology (EG-CLIMET), which was
a recent Cooperation in Science and Technology (COST) action financed by the European Union, noted that
there are hundreds of ceilometers deployed over Europe, which are currently under-exploited. EG-CLIMET
recommended that the ceilometers be networked to provide users easy access to calibrated backscatter data
(Illingworth et al., 2015). At the time of writing, profiles from 200 ceilometers from 17 countries are being
distributed in near real time by the E-Profile programme of European Meteorological Services Network
(EUMETNET, 2018) with the number expected to rise to about 700. The data formats, calibration techniques
and retrieval algorithms are being developed by COST action 1303: Towards operational ground based profiling
with ceilometers, Doppler lidars and microwave radiometers for improving weather forecasts
(http://www.toprof.imaa.cnr.it).

If ceilometer data are to be used in an operational context, and potentially for data assimilation, accurate
calibration is essential. The World Meteorological Organisation requirements (OSCAR, 2018) suggest the goal
for ice water content (IWC) observations is to have an accuracy of 10% and for aerosol optical extinction to
have an absolute accuracy of 0.01 km$^{-1}$, but no fractional accuracy is quoted.  Ice particle density is usually
assumed to be inversely proportional to particle size, so IWC is proportional to extinction and, for a given lidar
ratio, the requirement is for a ceilometer calibration accuracy to be 10%.

The use of theoretical calibrations for lidars and radars based on an accurate budget of the losses and gains in
the transmission and reception optics and in the electronics together with atmospheric attenuation can cause
large errors (Protat et al., 2011). Accordingly, it is preferable to find some natural target that has a known
backscatter value. There are two such candidates for ceilometers: firstly, the backscatter from the molecules in
the atmosphere and, secondly, the integrated backscatter profile from water clouds that totally extinguish the
lidar beam. In this paper, we will focus on the second method. This method, using the attenuated backscatter
signal from liquid water clouds, relies on the fact that the backscatter to extinction ratio (S) is a known value of
18.8 sr for wavelengths of relevance to ceilometers (O'Connor et al., 2004). The advantage of this method is





that the backscatter values from liquid water clouds are very high (typically peaking at 0.3 km$^{-1}$ sr$^{-1}$) so the signal to noise ratio of water cloud returns is very large. By contrast, the molecular signal close to the ground is over one hundred times lower than the cloud returns and of the order 10$^{-3}$ km$^{-1}$ sr$^{-1}$, falling off exponentially with height. For an accurate estimate of the molecular return it is necessary to average the ceilometer profiles over

several hours on selected cloudless nights when there is negligible backscatter from thin cirrus clouds or aerosols (e.g. Tsaknakis et al., 2011; Wiegner et al., 2014).

In this paper, we present a development of the calibration technique using liquid clouds that can be implemented operationally and which avoids the aforementioned potential problems. We report on the values of the calibration for the Met Office network of ceilometers and show the calibration stability in time. In section 2, we

review the specifications and performance of the two ceilometer models in widespread use in Europe. The calibration algorithm is described in section 3 and the instrument model dependent corrections and calibration results are addressed in sections 4 and 5. Finally, in section 6, we report on collocated ceilometer comparisons and statistics of the stability and accuracy of the calibration.

## 2 Instrumentation

### 2.1 The Met Office Ceilometer Network

Figure 1 shows the locations of the 40 ceilometers in the UK that are presently reporting the full vertical profiles of the attenuated atmospheric backscatter, and are referred to in this paper as the 'Met Office ceilometer network.' The purple crosses show the location of the 29 Vaisala CL31 ceilometers and the red circles show the 11 Jenoptik CHM15k Nimbus ceilometers that have been used to test the ceilometer calibration technique. Nine

sites have collocated Vaisala and Jenoptik ceilometers. Other Met Office ceilometers, many of which are the Vaisala CT25K model, report only cloud base height and are not discussed here, although the calibration technique can be applied to both the CT25K and the newer CL51 models. Note that Jenoptik no longer produce ceilometers; the manufacturing of them has been taken over by Lufft. From here on, we refer to these ceilometers as Lufft ceilometers, including those manufactured before production passed from Jenoptik to Lufft.

### 2.2 Vaisala CL31 ceilometers

The key technical properties of the ceilometers used by the Met Office are summarised in Table 1. In brief, the Vaisala CL31 ceilometers use an InGaAs diode laser which emits pulses with an energy of 1.2 μJ at a pulse repetition frequency (prf) of 10 kHz with a central wavelength of 910 ± 10nm, though the typical spectral width is more often 4 nm (Kotthaus et al. 2016, Markowicz et al. 2007). At these wavelengths, attenuation by water

vapour is significant, a fact overlooked by O'Connor et al. (2004). Due to the low power of ceilometers, they have a much higher pulse repetition rate compared to high-power lidars, to compensate for this lower power and to increase the signal to noise ratio. The CL31s have a single lens design, with the centre of the lens collimating the laser beam and the outer part of the lens used for focussing the backscattered light onto the receiver, which uses an avalanche photo diode (APD) detector to process the signal (Münkel et al., 2007). Complete overlap of

the transmitted beam at the receiver sample is achieved at a height of approximately 70 m (Martuccci et al., 2010) and the maximum range is 7.6 km.





| Specification | Vaisala CL31 | Lufft CHM15k |
|---|---|---|
| Laser | InGaAs Diode | Nd:YAG |
| Centre Wavelength | 910 nm | 1064 nm |
| Wavelength Variability | ± 10 nm | insignificant |
| Optical Design | Coaxial | biaxial |
| Pulse Energy | 1.2 μJ | 8 μJ |
| Pulse Repetition Rate (PRF) | 10 kHz | 5-7 kHz |
| Temporal Resolution | 30 s | 30 s |
| Vertical Resolution | 20 m * | 15 m |
| Complete Overlap | 70 m | 1000 m |
| Maximum Detection Range | 7.6 km | 15 km |

Table 1: Summary of some technical characteristics and parameters of the Vaisala CL31 and Lufft CHM15k, as operated in the Met Office network. * Exeter CL31 has a vertical resolution of 10 m.

There are currently several different versions of the firmware in use by the Met Office ceilometer network. The various versions process the signal in different ways, applying "cosmetic" shifts to the data to avoid unphysical negative backscatter values. The original users for ceilometer data were airline pilots and these cosmetic shifts were applied so that it was easier for non-experts to interpret the displays. Full details of the shifts and methods for correcting can be found in Kotthaus et al. (2016). These effects should certainly be corrected for in the study of smaller particles such as aerosols and ash; however, for the stronger signal from cloud particles the effect of these shifts on the calibration method shown here is negligible.

**2.3 Lufft CHM15k Nimbus ceilometers**

The Lufft ceilometers use a Nd:YAG laser and operate at a slightly longer wavelength of 1064 nm where the attenuation by water vapour is negligible. The APD detector employs a photon counting method. Due to the biaxial design of the Lufft ceilometers, full overlap is not reached until 1 km rather than 70 m for the CL31. The pulse repetition frequency is in the range 5 -7 kHz and the pulse energy is 8 μJ, which is six times higher than the Vaisala CL31 ceilometers. This higher pulse energy, combined with the different overlap configuration, results in a much higher sensitivity of the CHM15k ceilometer, for detection of elevated aerosols such as volcanic ash plumes.

**3 The calibration algorithm**

Autocalibration of ceilometers using liquid water cloud was proposed by O'Connor et al. (2004) as a simple method that requires no additional instruments to compute a calibration coefficient. The technique relies on the use of the lidar ratio (ratio of extinction to backscatter, denoted S) that is a constant for the droplets in liquid water cloud. Several studies have derived S from Mie theory: Pinnick et al. (1983) found that, for a wavelength of 1064 nm, S = 18.2 sr; Wu et al. (2011) calculated an S of 18.5 ± 0.47 sr for a wavelength of 1064 nm. O'Connor et al. (2004) calculated an S of 18.8±0.8 sr for a wavelength of 905 nm and showed that this was essentially constant for the observed cloud droplet size distribution for a mean droplet size ranging from 10 to 100 μm, but S values were lower for drizzle having larger droplets. Since S is very similar at 905 nm and 1064 nm, we follow O'Connor et al. and use S = 18.8 sr for both wavelengths.





The method compares this theoretical S to a calculated 'apparent S'. When the ceilometer signal is completely
extinguished by the cloud, the total path integrated attenuated backscatter $B$ is equal to the reciprocal of twice
the lidar ratio:

$$B = \int_{0}^{\infty} \beta_{observed} \, dz = \int \beta_{True}(z) \, exp[-2\tau(z)] \, dz$$

$$= \frac{1}{\eta S} \int exp(-2\tau) \, d\tau = \frac{1}{2\eta S} \qquad (1)$$

where B is the total integrated attenuated backscatter, $\tau$ is the optical thickness, S is the theoretical lidar ratio,
and $\eta$ is a multiple scattering correction which is dependent on laser wavelength, beam divergence, telescope
field of view, and altitude (z). The multiple scattering corrections are height dependent and calculated for each
gate using the fast method and code described by Hogan (2006; code available to download at
http://www.met.reading.ac.uk/clouds/multiscatter/). $\eta$ is usually between 0.7 and 0.85 for wavelengths between
905-1064 nm in liquid water clouds. The calibration technique involves multiplying the observed backscatter
signal $\beta_{observed}$ by a calibration coefficient, C, until $B\eta = 0.0266$ m$^{-1}$, the value for water drops when S =18.8 sr.
Note that C is a scaling factor, and is the reciprocal of the widely used "Calibration Constant" $C_L$ that is often
used for photon counting receivers, and is the factor by which the count should be divided to obtain a calibrated
value (e.g. Wiegner et al., 2014).

The calibration technique will fail if there are targets contributing to B that have an S that is not equal to 18.8 sr.
At ceilometer wavelengths, aerosols generally have S values above those for cloud droplets; marine aerosols
have an S close to 20 sr, but most aerosols have values that are much higher and in the range 40 to 100 sr for
dust, smoke and ash (e.g. Omar et al., 2009). If aerosols with S higher than 18.8 sr are included in profiles
leading to total attenuation of the signal, then the value of B will be less than for cloud alone, and the apparent
value of the calibration coefficient, C, will be too high. Conversely, drizzle has S values below those for cloud
droplets, so if drizzle is included in the profile, the value of B will be higher than for cloud alone, and the value
of C would be too low. The magnitude of the error due to aerosol depends on its optical depth beneath the cloud
layer; therefore, we can circumvent this uncertainty by not selecting profiles which have large backscatter from
aerosol. The inclusion of profiles that are not totally extinguishing the ceilometer return will also lead to values
of C that are too high, as will occasions when the window transmission is reduced or the pulse energy falls.

Figure 2a shows an example of an uncalibrated attenuating backscatter profile typical of those from
stratocumulus clouds that is ideal for use in the liquid cloud calibration algorithm. Cloud is observed as the
sharp peak in attenuated backscatter just above cloud base rising to a maximum value of 0.28 km$^{-1}$ sr$^{-1}$ within a
few range gates and clearly dominates the observed ceilometer return. The shaded area indicates the area of
integration used in computing the total attenuated backscatter of the profile. The profile in Fig. 2b is for a
stratocumulus cloud that completely attenuates the ceilometer return. However, in this case, it is unsuitable for
calibration because there is a significant return from aerosol in the lowest 200 m of the profile, and a more
gradual increase in attenuated backscatter below the peak at cloud base indicating the presence of drizzle below
the cloud.



A new algorithm has been designed to automatically sift through all profiles of attenuated backscatter, selecting

only those suitable for the cloud calibration according to a strict set of criteria. The method is fairly simple ensuring that it can be applied operationally with minimal impact on processing time. No absolute values of $\beta_{att}$ are required by the algorithm to evaluate the criteria below, so the instrument can be completely uncalibrated, or the calibration currently applied can have a large error. The algorithm only requires a minimum of 10 suitable profiles in a day for a calibration coefficient to be calculated. This means the calibration algorithm is suitable for

ceilometers at sites where liquid water cloud can be sparse and infrequent. There are two main sets of criteria that must be met by the profile of attenuated backscatter for it to be used to calculate a calibration coefficient:

1. Unsuitable individual profiles:

    a. Aerosol filter. In any single profile, if the aerosol under the cloud contributes more than 5% to the total integrated backscatter (as shown in Fig. 2b), then this profile is removed from the
calibration. The transmission through the aerosol below the cloud attenuates the ceilometer beam and this attenuation increases with greater concentrations of aerosol. If the aerosol has a lidar ratio value twice the value assumed for cloud droplets, then this filter should limit the calibration error to a maximum of 5%.

    b. Peak sharpness filter. The peak backscatter magnitude must be a factor of 20 greater than the
value 300 m above and below that peak. A liquid water cloud suitable for calibration must fully attenuate the ceilometer beam; therefore, the backscatter values should decrease rapidly in the gates immediately above the peak value. Additionally, drizzle or rain below the cloud may give a large backscatter signal and, like the aerosol, will distort the apparent lidar ratio. Hogan et al. (2003) report that individual liquid-water layers do not tend to occupy more than
300 m of the ceilometer profile, due to their strong attenuation. Our own observations of the data lead to the same conclusion. This filter should therefore remove profiles that do not fully attenuate the beam and those that contain drizzle or rain.

    c. Window transmission and pulse energy check. A check is made on the recorded instrument transmission (given as a percentage of how much of the instrument window is clear) and on
the reported pulse energy (given as a percentage of a nominal amount). Both of these conditions can affect the true value of attenuated backscatter. For considering instrument and calibration stability, periods affected by reduced window transmission and/or reduced pulse energy are filtered out at a threshold of 90% or less.

2. Consistency of neighbouring profiles

    a. Lidar ratio stability – This filter traps errors due to patchy cloud cover or drizzle that may not have been identified by the first filter by checking that the apparent lidar ratio is the same as its nearest neighbours. The recommendation is to compare to 3 profiles either side; however, if the ceilometer is at a site where liquid water cloud is infrequent, this could be reduced to 1
or 2 profiles either side, with consequent degradation of the accuracy of the calibration coefficient. There must be at least 10 acceptable profiles for a calibration coefficint to be recorded for that day.



The operation of these filtering procedures in removing unsuitable profiles is illustrated in Fig. 3a, where a
stratocumulus cloud layer located at about 1 km for the whole day is ideal for calibration. The liquid cloud
backscatter signal, which has values greater than $10^{-0.5}$ km$^{-1}$ sr$^{-1}$ (in the red region of the colour scale), appears as
a thin layer above which there is only noise. Within the noise, the diurnal cycle of the skylight is visible. The
noise in the data is visible as speckling, and is of the magnitude of less than $10^{-3}$ km$^{-1}$ sr$^{-1}$. There are also limited
periods of broken, patchy cloud, which are identifiable by breaks in the layer of high backscatter, and limited
periods of drizzle which are identifiable by the fall streaks (cyan colours of the order $10^{-2.5}$ km$^{-1}$ sr$^{-1}$) below the
cloud.

Figure 3b illustrates the two main filtering steps of the new calibration algorithm. The thick, light grey line
shows the apparent S values for each individual profile that is acceptable and has removed those profiles
between from 3.30 – 4.30 hours where there is drizzle and aerosol below the cloud base, but this still leaves
some large apparent S values from 12.00 -15.00 hours that are due to broken cloud that does not totally
extinguish the ceilometer return. The second filter checks for consistency between neighbouring profiles and
successfully identifes and removes these spurious profiles where there is broken cloud.

The remaining profiles (i.e. those in black in Fig. 3b) give the values of apparent S which would be used to
calculate C using Eq (1). It is evident that these values remain very constant over the course of the day, implying
that the calibration of the instrument is very stable on this time scale. This is important, since our method can
only be applied during cloudy conditions, which may be separated by intervals of several days. The stability of
C implies we can interpolate between calibration events.

**4 Calibration of 910 nm ceilometers**

**4.1 Water vapour attenuation**

To complete the calibration of the Vaisala CL31 ceilometers (and others of similar wavelength – e.g. Vaisala
CL51, CT25k, CT75k, Campbell Scientific CS135), the effect of atmospheric water vapour below the cloud on
the laser signal must be considered. This is because the wavelength of these ceilometers (910 nm) is in a weak
water vapour absorption band. Note that because the Lufft ceilometers operate at 1064 nm, where there is a
water vapour absorption window, those ceilometers do not require a correction.

A recent paper by Wiegner and Gasteiger (2015) describes a method of correcting for water vapour attenuation
for ceilometers at wavelengths around 910 nm by performing detailed line by line radiation transfer calculations
and investigating the impact of the instrument emission spectrum (e.g. Vaisala states that for a CL31 the
wavelength is $910 \pm 10$ nm at 25$^0$C and with a drift of 0.3 nm K$^{-1}$). As the housing of the ceilometer lasers and
detectors are temperature-controlled environments, the effect of laser wavelength drift due to temperature can be
considered insignificant. However, even if the potential for drift is ignored, Wiegner and Gasteiger's method
still requires the use of a radiative transfer model or access to their "WAPL" database of absorption coefficients.
Because the liquid cloud calibration method presented in this paper is intended for operational, real-time use, a
simple, robust and computationally cheap method was required.





A simplified technique for correcting for the two-way water vapour attenation has therefore been devised based
on Markowicz et al. (2008) who show that the normalised spectrum of laser emission is wide enough to smooth
out  the individual water absorption lines so that for a water vapour path of 2 cm, a typical summer value in the
UK,  the change in water vapour transmission varies from about 0.77 to 0.75 (about 3%) as the peak laser
emissivity increases from 900 to 916 nm. The typical water vapour path in winter is 1cm leading to a
transmission of about 0.85, so if no water vapour correction was made, one would expect an apparent annual
cycle of the calibration coefficient of about 12%. The water vapour could be estimated using a microwave
radiometer. Alternatively, it can be obtained from a numerical weather prediction (NWP) model. In this paper
we take the latter approach. Cossu et al. (2015) have compared NWP output with the water vapour path derived
from microwave radiometers and find that the mean bias of the NWP water vapour path is only 0.7mm.

A simple monotonic function has been fitted to data from Markowicz et al. (2008) in order to parameterise the
two way attenuation by water vapour as a function of integrated water vapour (IWV) up to 2cm at wavelengths
between of ~910 nm depicted in Fig. 4:

$$T_{wv} = 1 - 0.17 IWV(z)^{0.52} \tag{3}$$

where $T_{wv}$ is the two way transmission as a percentage of the transmission without water vapour attenuation and
IWV(z) is the atmospheric water vapour content from the surface to height z in g cm$^{-2}$. The attenuated
backscatter is then corrected using:

$$B = \int \beta_{att} \times C_{wv} \; dr, \text{ where } C_{wv} = \frac{1}{T_{wv}}. \tag{4}$$

The transmission calculation for each range gate requires the water vapour content obtained by integrating the
water vapour density from the ground to each specific range gate, resulting in a transmission profile. For the
automatic operational calibration of the Met Office ceilometers, water vapour density would be calculated from
the Met Office UKV model, a convection-permitting variable resolution regional model run operationally over
the UK (Tang et al., 2012), using pressure, temperature and specific humidity. A comparison of the detailed
line-by-line Wiegner and Gasteiger method with the simpler approach using the water vapour density profiles
obtained from the ECMWF model provided by Maxime Hervo (Meteoswiss, Personal Communication, 2016) is
shown in Fig. 5.  The WAPL method is depicted in blue and the new, simple method is in red. The
transmissivity profiles differ by a maximum of 2% for a total transmissivity of 0.85.

### 4.2 Region of integration

For the Vaisala ceilometers in the Met Office network, a cosmetic feature in the firmware suppresses the range
correction to the received power for heights above 2.4 km, except when there are clouds present.  This is done to
avoid the background noise signal leading to apparent clouds at high altitudes that confuse the non-expert. In
addition, Kotthaus et al. (2016) found that the attenuated backscatter in the lowest 200 m may be subject to
artefacts so, in this calibration study of the Met Office's Vaisala ceilometers the cloud returns above 2.4 km and
below 200 m are not used.

Figure 6 shows a histogram of the integrated attenuated backscatter, B, from liquid cloud as a function of the
height of the maximum attenuated backscatter (used as an indicator of the height of the cloud), for profiles from





an uncalibrated Met Office Vaisala CL31 situated at Middle Wallop (51.15° N, 1.57° W). Multiple scattering
and water vapour attenuation below the cloud have been accounted for. Over 100,000 profiles were used, from
the period September 2014 to December 2015. The numbers superimposed on the right side of the plot show the
mean and standard deviation at 100m intervals of the range. For 16 of the 21 heights shown, the mean value is
0.021 sr$^{-1}$. The other 5 gates differ by a maximum of only 0.002 sr$^{-1}$. This provides confirmation both of the
validity of the range dependent multiple scattering correction and the assumption of constant S for different
water clouds.

Below 500 m there is, however, a slight change. The distribution of the integrated attenuated backscatter is still
concentrated in a similar region as at other heights, but it also has a slight tail to the left. For profiles in this tail
region below 500 m, the attenuated backscatter is smaller, which will result in a larger apparent lidar ratio. The
mean value of the integrated attenuated backscatter at heights below 500 m decreases by 9.5% with the standard
deviation increasing by 17%. We suspect that this is a result of the instrument detector saturating. When the
cloud is very low, the cloud signal may be so strong at its peak that the true magnitude of the backscattered
signal is not fully detected and, therefore, the integrated attenuated backscatter appears smaller when compared
to other heights. It is also possible that this may, occasionally, be due to microphysical processes within the
cloud. Nicholls (1984; Powlowska et al., 2000) showed that there is a reduction in droplet number concentration
below 450-500 m. This may, in some cases, be significant enough to affect the backscatter at this height.
Therefore we also reject profiles where the cloud is between 200 – 500 m.

**4.3 Calibration results for Middle Wallop**

Figure 7 shows a time series of the calibration results for the CL31 at Middle Wallop in Southern England over
a period of 20 months. The top panel shows the mode of the calibration coefficient C, for each day with
sufficient (minimum 10) attenuated backscatter profiles deemed suitable by the calibration algorithm. For
example, the black cross on 25 October 2014 is the mode of the calibration coefficients calculated from the
filtered S values (per profile) shown (in black) in Fig. 3b.

The results are for almost two years of data and establish that the calibration remains stable over time. The
number of profiles used for the calculation of the daily value is different depending on the occurrence of cloud
on each day. As the calibration algorithm requires only a minimum of 10 profiles to be included in the daily
value of the mode, even a short period of cloud will be included for calibration purposes. This ensures that the
technique can be applied to ceilometers in locations with climates that have relatively little cloud occurrence.
For this site, the algorithm found a minimum of 8 days every month with profiles suitable for calibration, with
slightly more suitable days during autumn and winter. The water vapour correction profiles are calculated from
the Met Office UKV model at the grid point over the Chilbolton Observatory, which is approximately 15 km
from Middle Wallop. The variables needed to calculate the transmission profiles were available every hour, and
have been interpolated to the observational time.

The middle panel of Fig. 7 shows the daily mean and standard deviation for the same station and data. For the
20-month (574 days of data available) period, a calibration was possible on 320 days, or 56% of the days, and
the average number of profiles per day was 292. There were just 7 days out of the 320 when the calibration
coefficient, C, was approaching 2.0 rather than the median value of 1.4, so a 90-day running mean was



calculated and is displayed in the lower panel of Fig. 7. This running mean had a value of C of 1.40 with a standard deviation of 0.021; this is less than 2% of the mean. As both the mean height of the cloud base (and

therefore the amount of multiple scattering) and the water vapour attenuation have a pronounced annual cycle, this low value of standard deviation is evidence of the appropriateness of the algorithms that correct for these two effects. Accordingly, it is recommended that for automatic, operational use for a ceilometer, without window transmission or pulse energy issues, a 3-month running average of the calibration coefficient be used.

### 4.4 Calibration results for the Met Office network

The calibration of all the Vaisala CL31 ceilometers in the Met Office network has been collated and is summarised in Fig. 8, where box and whisker plots are shown of the calibration coefficient for each of the CL31s, calculated from data for the period January – March 2015. All instruments have a calibration coefficient larger than 1.0, with the majority of the instruments having a coefficient of around 1.5. The range of coefficients for each station is small, with 50% of the data (contained within the box) being within 10% of the mean value.

The colour code in Fig. 8 indicates the different firmware versions installed on the instruments within the Met Office ceilometer network. Stations using the 202 firmware, which are shaded pink (for example, Aberporth, Coningsby, Middle Wallop), tend to have an even smaller range of C values, with 50% of the data being within 8% or less of the mean. The network includes stations from Lerwick (60.16°N, 1.15°W) down to Gibraltar (36.14°N, 5.35°W), demonstrating that the calibration method has been successfully applied to a range of

different climates, from the North Sea down to the Mediterranean Sea.

The water vapour correction of the data has been applied for the calibrations depicted in Fig. 8, as described in section 4.2. Ideally, the water vapour profiles for each specific site should be used to calculate the transmission correction. Due to data availability, only the model data for Chilbolton was available at this time. As the calibration specifically requires a cloud base below 2.4 km and the air is generally well-mixed below cloud base,

the water vapour path mixing is generally fairly constant and depends on the temperature and height of the cloud base. Therefore, it is assumed that the season is more important than the location and so the same water vapour profiles are used for all the ceilometers. In future, operational implementation, the site specific vapour profile would be used.

Figure 9 shows the calibration of the Gibraltar CL31 ceilometer in more detail and has the same format as Fig.

7, for 12 months at Gibraltar rather than the 20 months at Middle Wallop. As the UKV does not cover Gibraltar, the water vapour correction was calculated using data from the Met Office Unified Model. Due to the climate, the number of occasions when there are suitable clouds for calibration is reduced at the Gibraltar site. In one year there were 51 days of suitable clouds, with each day having on average 128 profiles. However, from mid-May to mid-Sept. there were only two days where calibration was possible and in December there were none.

While this is in part due to a lower amount of stratocumulus compared to the UK, it was also caused by the window transmission. The Gibraltar ceilometer requires regular cleaning as the dust tends to build up on the window, reducing the transmission. Therefore several days where the window transmission dropped below 90 % have been filtered out by the algorithm. The 4 crosses in Fig. 9a and 9b which show a calibration coefficient closer to 2.0 correspond to days where the profiles only just pass the 90 % window transmission check.

Nevertheless, Fig. 9c confirms that the 90-day running mean calibration coefficient over the twelve month



period was 1.5 with a standard deviation of 0.05, or about 3%, and, as with the data in Fig. 7, there is no sign of an annual cycle in the calibration coefficient.

## 5 Calibration of 1064 nm ceilometers

We now address the issue of cloud calibration for the Lufft ceilometers, which operate at a wavelength of 1064 nm. It should be noted that many high power lidars have a channel at 1064 nm and can also be calibrated with the liquid cloud method. However, as they do not have the same firmware and hardware issues as ceilometers, high power lidars are not directly discussed here.

### 5.1 Saturation issue

Before the cloud calibration can be applied to the Lufft ceilometers, the issue of saturation must be addressed.
Due to the greater pulse energy (compared to Vaisala ceilometers), the Lufft ceilometer receivers are much more prone to saturation. When saturation occurs, the backscatter reported for this profile is false – it is too low. Hence, these profiles that saturate need to be avoided. The exact magnitude of power at which the Lufft power saturates is unknown. However, it is possible to detect the majority of saturated profiles, because the saturation of the receiver usually causes the output to overshoot to an unphysical negative value just above the cloud echo
(Personal Communication, H. Wille, Lufft, 2017).

The first panel of Fig. 10 demonstrates the impact of saturation and the subsequent negative overshoot: the blue profile, from the lower cloud base where saturation has occurred, has a smaller magnitude than the red profile of the higher cloud that has not saturated. If a saturated profile were to be used for calibration, then the total attenuated backscatter recorded by the ceilometer would appear lower than a non-saturated profile and would,
therefore, systematically skew the calibration coefficient to be larger than it should be.

Because the profiles that saturate have this apparent layer of negative attenuated backscatter, this can be used as a check in the calibration algorithm to remove these profiles. There is a correlation between the negative backscatter and the magnitude of saturation: the larger the negative backscatter value, the greater the magnitude of saturation, but this relationship is not linear and so the saturation cannot be easily corrected (Personal
Communication, H. Wille, Lufft, 2017). Hence, in what follows, we simply filter out such profiles completely. To ensure it is the negative attenuated backscatter of a saturated profile that is detected and not just the random noise in the profile above the cloud (which appears as small positive and negative values varying randomly from gate to gate), any profiles which have a layer of negative backscatter greater than 100 m are removed from the calibration.

This method removes the majority of saturated profiles; however, those profiles which only just saturate the instrument receiver may not always result in a region of negative attenuated backscatter. To increase confidence that all saturated profiles are being removed, a cloud height threshold can be imposed. Figure 11 shows a histogram of the uncalibrated integrated attenuated backscatter for profiles in liquid water cloud at Aberporth. The multiple scattering correction has been applied, but profiles where the instrument saturates have not been
filtered out. Therefore, one can see clearly the impact of saturation.





As shown for the Vaisala ceilometer calibration (Fig. 4), the integrated attenuated backscatter should be a constant, independent of the height. This is not the case for the Lufft ceilometer. This is because when the instrument saturates, the received power becomes limited to some (unknown) maximum power. Backscatter is proportional to received power $\times$ range$^2$ that means the integrated backscatter measured will appear to be a

function of range. This is in contrast to the Vaisala ceilometer in Fig. 6 where saturation is not occurring. Therefore, Fig. 11 shows that saturation is occurring below a height of 2.2 km because of the systematic change in backscatter with range. Above 2.2 km, the integrated backscatter does not change systematically with height, showing that these higher-level clouds are not saturating the ceilometer receiver – since they are further away, the received power is weaker and below the level where saturation occurs. The exact height where the integrated

attenuated backscatter becomes constant will be instrument-specific as it will be dependent on instrument power and on the individual receiver. However, with this simple test, the height threshold required can be easily found, thus allowing for the saturated profiles to be removed and calibration to be correctly calculated.

### 5.2 Calibrated results for Lufft instruments

The calibration algorithm can now be applied to the Lufft ceilometers in a way similar to the calibration of the

Vaisala ceilometers. A couple of changes are included. Unlike the Vaisala ceilometers, the Lufft ceilometers are not restricted by a change in processing at 2.4 km, so the upper range limit of integration to compute B is increased to 4 km, which incorporates the vast majority of liquid clouds in the UK. Additionally, the higher cloud range means the ceilometer beam must travel through a larger portion of the atmosphere, so the ratio filter (criteria 1a) is increased from 5% to 10%. Note that this may lead to a slightly larger uncertainty in C, but this is

done to decrease the amount of data that would otherwise be filtered out. The lower height limit is also changed, so that clouds below 1 km are not used. This is to avoid using profiles in the region where an overlap correction is applied as there is a potential temperature dependency in the overlap function that has not been accounted for (Hervo et al., 2016).

At the 1064 nm wavelength there is no absorption by the water vapour molecules; so no water vapour correction

is required. Figure 12 shows an example of the cloud calibration applied to a Lufft CHM15k ceilometer situated at Aberporth, West Wales, for the twelve months of 2015. Because of the requirement to remove the low level clouds that resulted in saturation, calibration was only possible on 70 days or about 20% of the days. Each day had an average of 58 profiles; nevertheless, the 90-day running mean calibration coefficient over the year was 0.48 with a standard deviation of 0.02 or 4%, with no sign of any annual cycle. This standard deviation of 4%

over the year is slightly higher than for the Vaisala ceilometers, probably because of the relaxation of the threshold for aerosol to be considered negligible, but is well within the specified requirement of 10%.

The calibration has been applied to the rest of the Lufft ceilometers in the Met Office ceilometer network, as shown in Fig. 13. Most of the sites have a relative calibration of less than 1.0; however Coningsby has a particularly large calibration coefficient. This highlights the importance and need for a calibration of each

instrument. For each site, the relative standard deviation is small.

### 6 Collocated comparisons



The majority (9 out of 11) of the Lufft ceilometers are collocated with a Vaisala ceilometer, allowing comparisons between the two types. Figure 14 compares the observations of attenuated backscatter from the two ceilometers at Aberporth, which have both been calibrated using the cloud method. To make a fair comparison between the two instruments, it is necessary to choose the meteorological situation carefully. Aerosols are problematic, because the ceilometers operate at different wavelengths, and the backscatter from aerosols is wavelength-dependent in a way that we do not know a-priori. We could analyse profiles in liquid clouds; however, we have already used these for calibration (so it would not be a truly independent test). In addition, the backscatter profile in liquid clouds contains very large gradients which make any comparison extremely sensitive to small offsets in range and/or differences in range-gating between the two instruments. Rain profiles could potentially be used for the comparison: however, rain which reaches the ground may wet the telescope optics and affect the data.

Better targets for such comparisons are drizzle drops and ice particles. These particles are large compared to the wavelength of the lidar, and hence the scattering is almost wavelength-independent (since we are close to the geometric optics regime). At the same time, the extinction of the lidar beam is much more gradual than in liquid cloud, providing smoothly varying backscatter profiles, which can be interpolated onto a common grid with little error. If we use an ice case, we would need to account for the influence of specular reflections from oriented ice crystals (e.g. Westbrook et al 2010a). Therefore in this example, a drizzle scene has been chosen.

To establish quantitatively whether the backscatter for drizzle drops at 1064 and 910nm are actually equal, Mie calculations were performed, assuming Gamma drop size distributions (Westbrook et al., 2010b). The results show that the backscatter at 1064nm is very similar to that at 910nm, but systematically smaller. The differences are very modest: between 5 and 8% for median drop diameter in the range 0.1-0.6mm, with most of the calculated values in this range close to 7%. Meanwhile the extinction is essentially identical for both wavelengths. Thus, if the calibration has been successful it would be expected that the backscatter profiles in drizzle would match very closely. However, the Lufft is systematically 7% smaller than the Vaisala, if no adjustment to account for the different wavelengths is made. Therefore, for this comparison, the Vaisala attenuated backscatter data have been reduced by 7%.

It is also necessary to consider the various technical issues already discussed earlier in the article when selecting profiles, in particular the need for the drizzle to be high enough to be in the fully overlapped region for the Lufft instrument, and below the 2.4km height above which the Vaisala ceilometer data range correction is variable. Therefore, the data used covers the period 00.00 to 15.00 GMT on 22nd April 2016, during which time there is drizzling cloud. The data are 10-minute averages of attenuated backscatter between 1.0 and 2.4 km and the Vaisala data has been regridded from 20 m resolution to 15 m resolution using linear interpolation. The quicklooks of the attenuated backscatter for the Vaisala and Lufft ceilometers are shown in Fig. 14a and Fig. 14b, respectively.

Figure 14c shows that there is a strong correlation close to the 1:1 line between the two ceilometers, as the intercept is close to zero and the slopes differ by less than 10%. The spread of the individual data points is rather larger than 10% and can be accounted for by the different resolutions and interpolation errors. This comparison of the two different types of ceilometers confirms the reliability of this calibration method – the two



independently calibrated ceilometers, each with their own challenges (e.g. water vapour, saturation), are consistent with each other. This result is important for an operational network such as the Met Office ceilometer network because it helps maintain a reliable, comparable stream of calibrated data, with water vapour and saturation successfully accounted for, from each instrument at each site.

**7 Conclusion**

In this paper, we have presented a robust algorithm to calibrate ceilometers based on the cloud calibration technique that relies on the fact that the lidar ratio of liquid water clouds is a known constant. This new method can be run operationally, removing unsuitable profiles where the cloud does not fully attenuate the ceilometer beam or where there is significant backscatter from aerosols. By excluding profiles when the low cloud leads to instrument saturation (particularly in the Lufft instruments) or when the window transmission is low, and by accounting for the attenuation of the ceilometer beam by water vapour (in the Vaisala instruments), we show that ceilometers from different manufacturers can be successfully calibrated using this method. It has been demonstrated that the running 90-day mean calibration coefficient for each instrument over a year is constant to better than 5% with no detectable annual cycle. At the time of writing, profiles from 200 ceilometers from 17 countries are being distributed in near real time by the E-Profile programme of European Meteorological Services Network (EUMETNET) with the number expected to rise to about 700. E-Profile has decided to calibrate the Vaisala ceilometers using the cloud calibration technique described in this paper.

*Author contributions:* EH wrote the paper. All authors contributed to the scientific ideas in the paper EH and CCP wrote the code to perform the calibration. CCP and SB provided EH with access to observational data. All authors discussed the results and edited the manuscript.

*Acknowledgements:* This study was jointly funded by a Natural Environment Reseach Council (NERC) PhD Studentship and the Met Office Industrial CASE Partnership. The authors would like to acknowledge the contribution of the COST Action ES1303 (TOPROF) and in particular Ewan O'Connor and Maxime Hervo for many helpful discussions. We would also like to thank Mariana Adam, Joelle Buxmann and Jacqueline Sugier (Met Office) for their help and support during this PhD project.

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





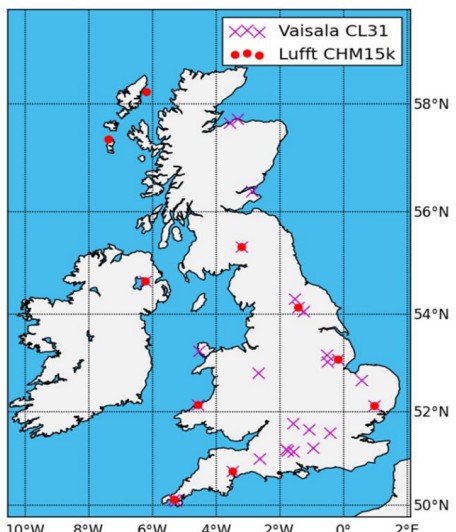

Figure 1: Location of Met Office ceilometers which record the full profile of attenuated backscatter. The Vaisala CL31s are indicated by a purple cross and the Lufft CHM15k by red dots.

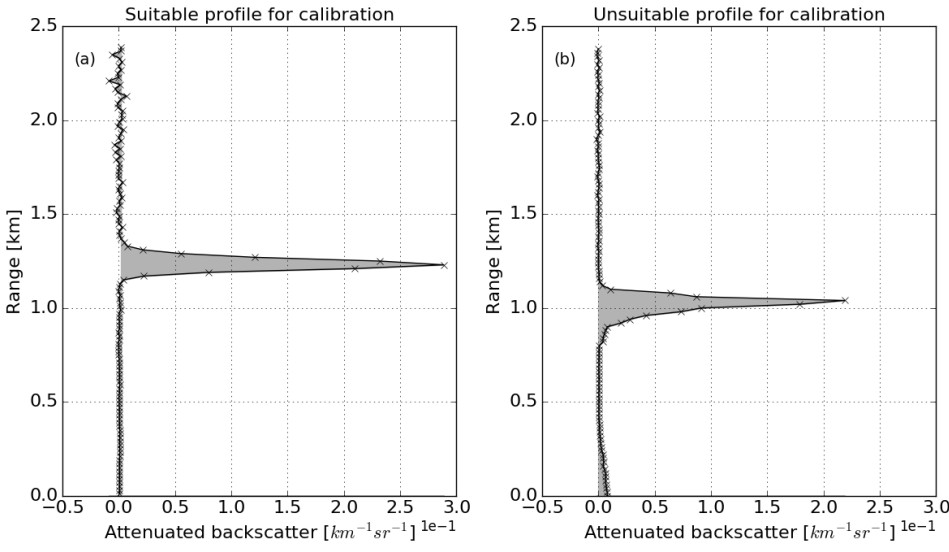

Figure 2: Profiles of attenuated backscatter through stratocumulus cloud. Panel (a) shows an example of a suitable profile for calibration. The integral of the profile (grey shaded area) is equal to $\frac{1}{2\eta S}$ and, when calibrated, should give an S of $18.8 \pm 0.8$ sr. Panel (b) shows an example of a profile unsuitable for calibration due to the high levels of aerosol in the first 200 m, indicated by the grey shading up to 200 m, and due to the drizzle below the stratocumulus cloud, indicated by the slight increase in attenuated backscatter underneath the peak.




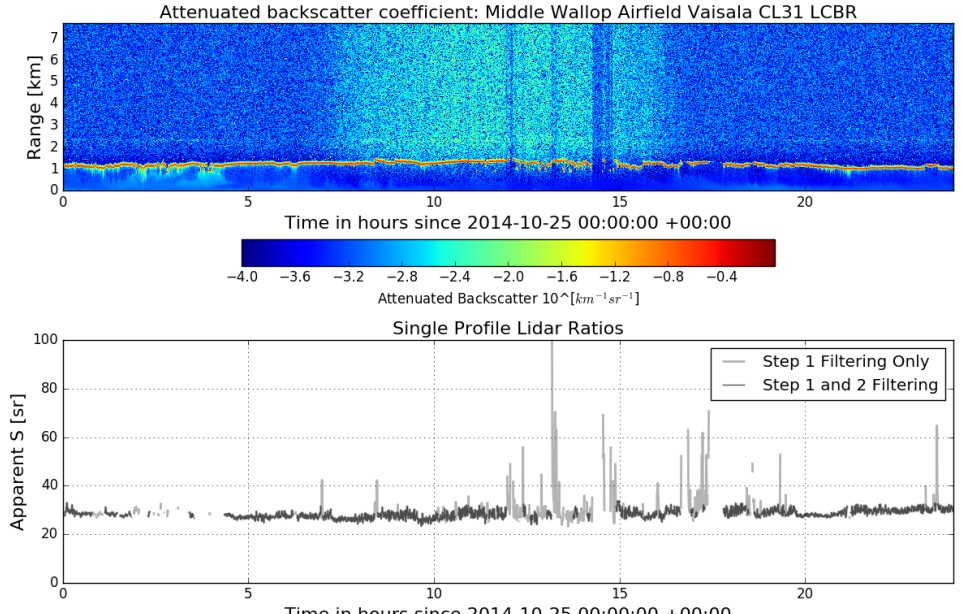

Figure 3: (a) Uncalibrated attenuated backscatter vertical profiles (colours shown on a log scale) for 25th October 2014 from a CL31 ceilometer at Middle Wallop airfield (51.1489 N, 1.5700 W) and (b) the apparent lidar ratio for the same day. In panel (b), the grey line shows the apparent S for profiles that pass the step 1 filtering of the calibration algorithm and the black line shows the profiles that pass the step 2 filtering and are used to calculate the calibration coefficient.




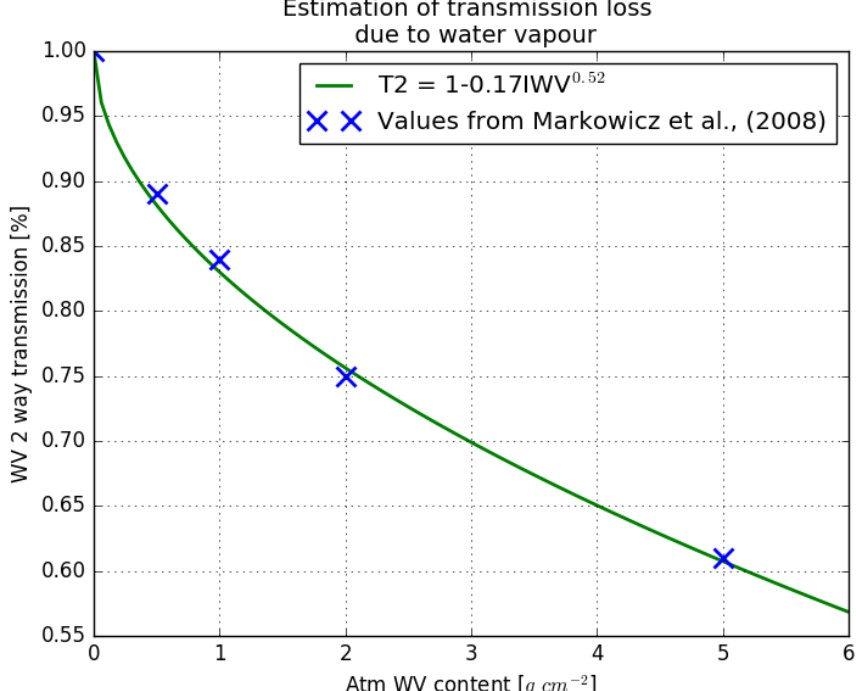

Figure 4: Estimated transmission loss due to the atmospheric water vapour content. The blue crosses are the
values calculated by Markowicz et al. (2008) for a ceilometer with a wavelength of 910nm.

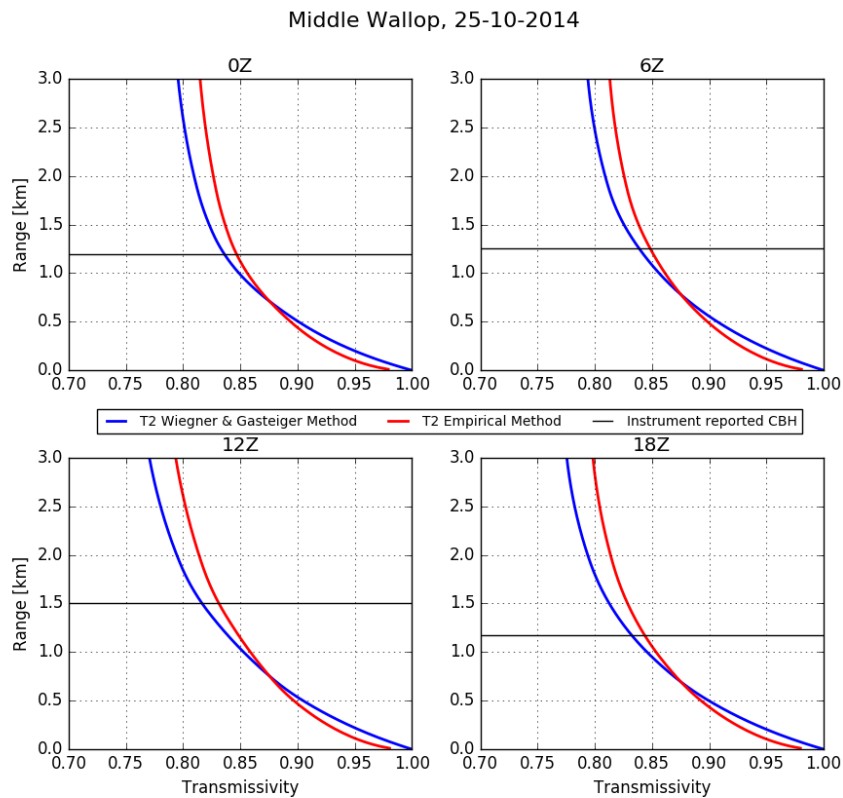

Figure 5: Comparison of water vapour transmission correction methods using ECMWF water vapour density profiles for 25th October 2014 at Middle Wallop, England. In blue, the transmissivity is calculated using WAPL (Wiegner and Gasteiger, 2015) and in red the transmissivity has been calculated using the empirical function shown in Fig. 4. The black lines show the instrument reported cloud base height at that time.





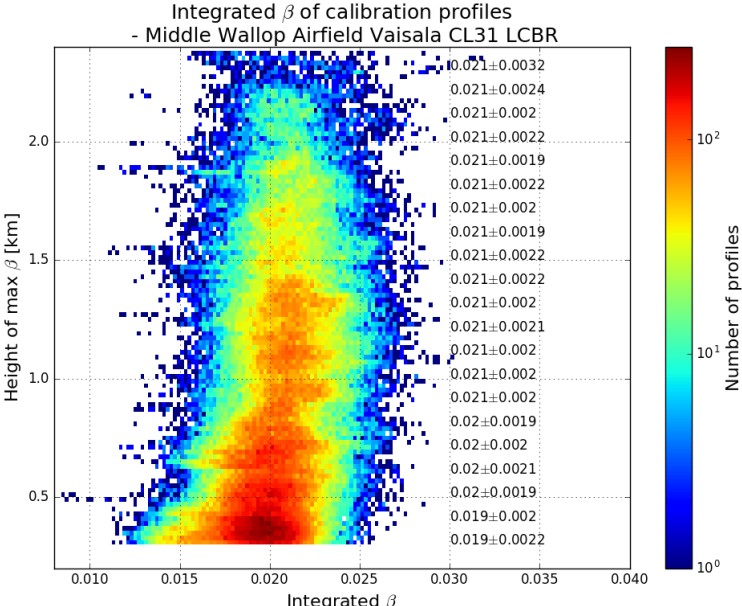

Figure 6: 2D histogram of integrated attenuated backscatter with range, with height dependent multiple scattering correction applied. Darker colours (towards red) indicate a higher density of profiles. The values shown along the right-hand side give the mean +/- the standard deviation of the integrated attenuated backscatter (units sr$^{-1}$) at 100m intervals.



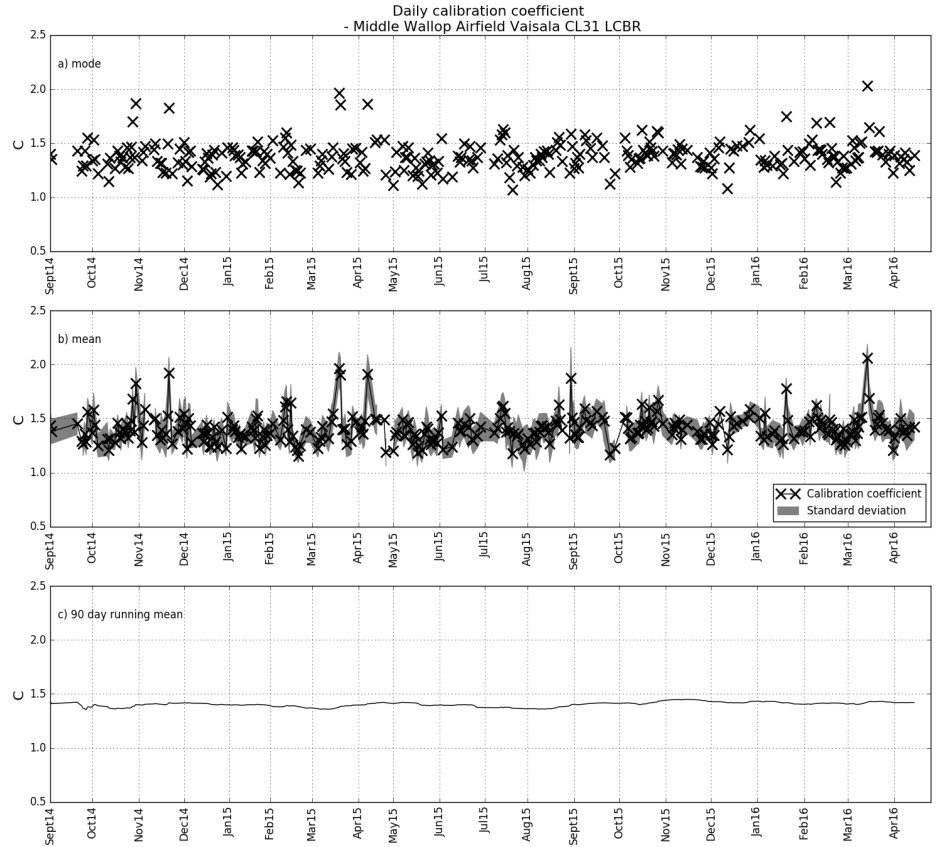

Figure 7: Calibration Coefficient (C) for Middle Wallop CL31 from September 2014-April 2016. Each black cross represents a single day, calculated from profiles deemed suitable by the calibration algorithm. Panel a) shows the mode of C for each individual day, b) shows the mean of C for each day, with the standard deviation shaded in grey and c) shows a 90 day running mean for the 20-month period. The average of the daily modes is 1.38 ± 0.14, the average of the daily means is 1.41 ± 0.13 and the average of the 90-day running means is 1.40 ± 590 0.021.



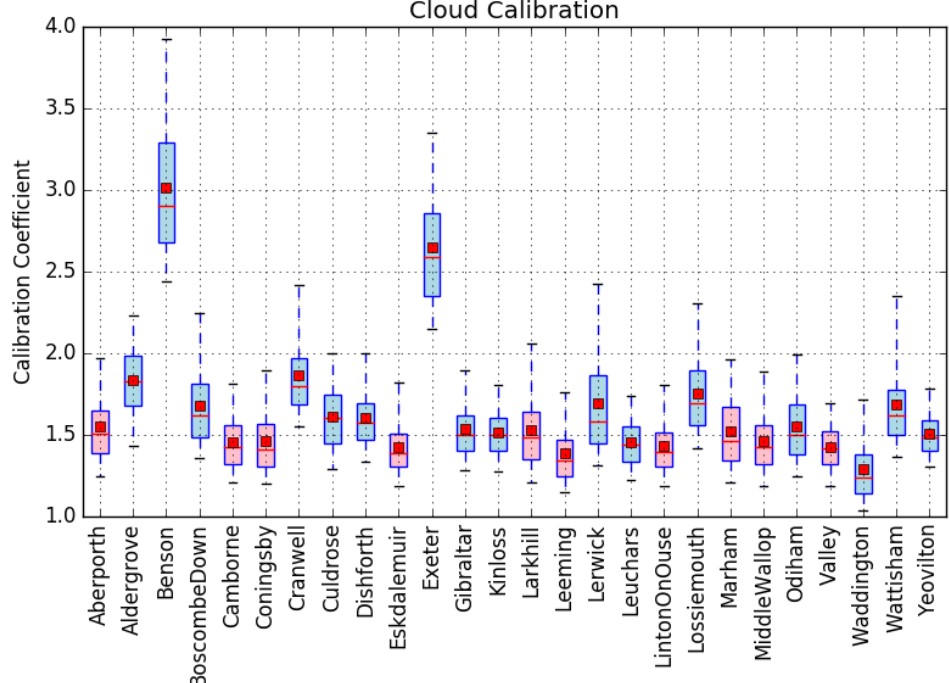

Figure 8: Calibration coefficient for each of the CL31 ceilometers in the UK Met Office network. 3 months of data (January-March 2015) have been used for each instrument. The number of suitable calibration profiles will be dependent on occurrences of cloud and, therefore, will vary for each instrument. The box outline represents 50 % of the calibration profiles and the whiskers extend to include 95% of the profiles (outliers have been 595 excluded from plot). The horizontal red line in the box shows the median calibration coefficient and the smaller, filled box shows the mean. The boxplot is shaded by firmware version as given by the ceilometer files on 1st January 2015: pink for version 202 and blue for versions 170 and 172.





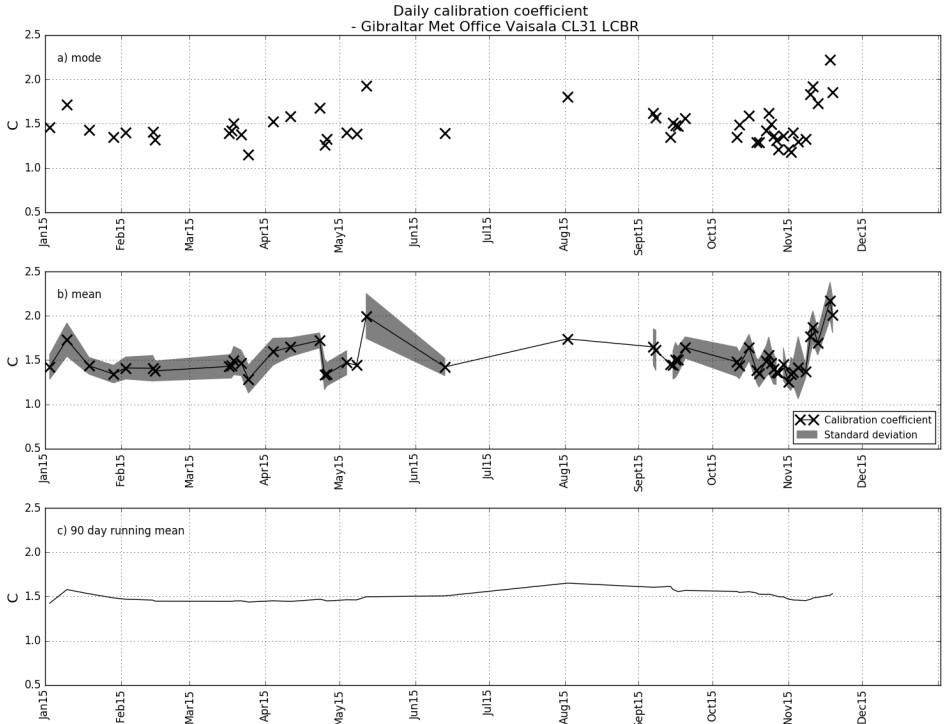

Figure 9: Calibration Coefficient (C) for Gibraltar CL31 from January-December 2015. Each black cross represents a single day, calculated from profiles deemed suitable by the calibration algorithm. Panel a) shows the mode of C for each individual day, b) shows the mean of C for each day, with the standard deviation shaded in grey and c) shows a 90 day running mean for the 12-month period. The average of the daily modes is 1.48 ± 0.21, the average of the daily means is 1.51 ± 0.19 and the average of the 90-day running means is 1.50 ± 0.053.

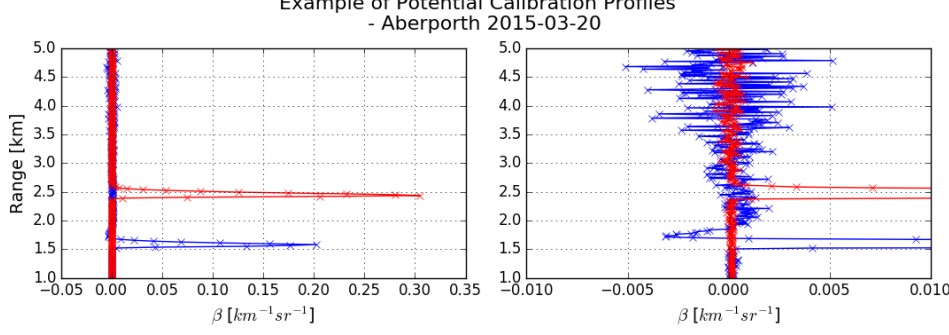

Figure 10: Two profiles of attenuated backscatter that detect liquid cloud from the Lufft CHM15k ceilometer at Aberporth on 20th March 2015 (second panel shows same plot on different scale). The profile in blue has a negative overshoot above the cloud, whereas the red profile does not.





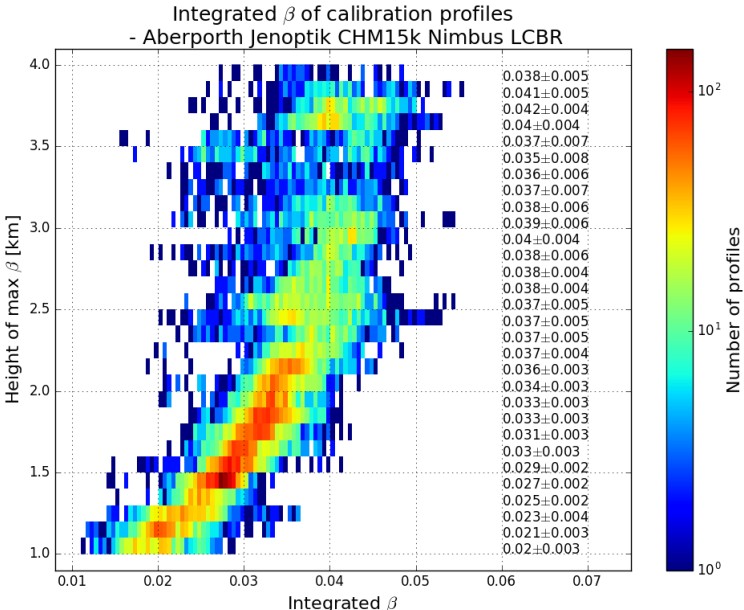

Figure 11: 2D histogram of the value of the integrated attenuated backscatter in profiles used for calibration with range. A height dependent multiple scattering correction has been applied. Darker colours (towards red) indicate a higher density of profiles. The values shown along the right-hand side give the mean +/- the standard deviation of the integrated attenuated backscatter (units sr$^{-1}$) at 100m intervals.




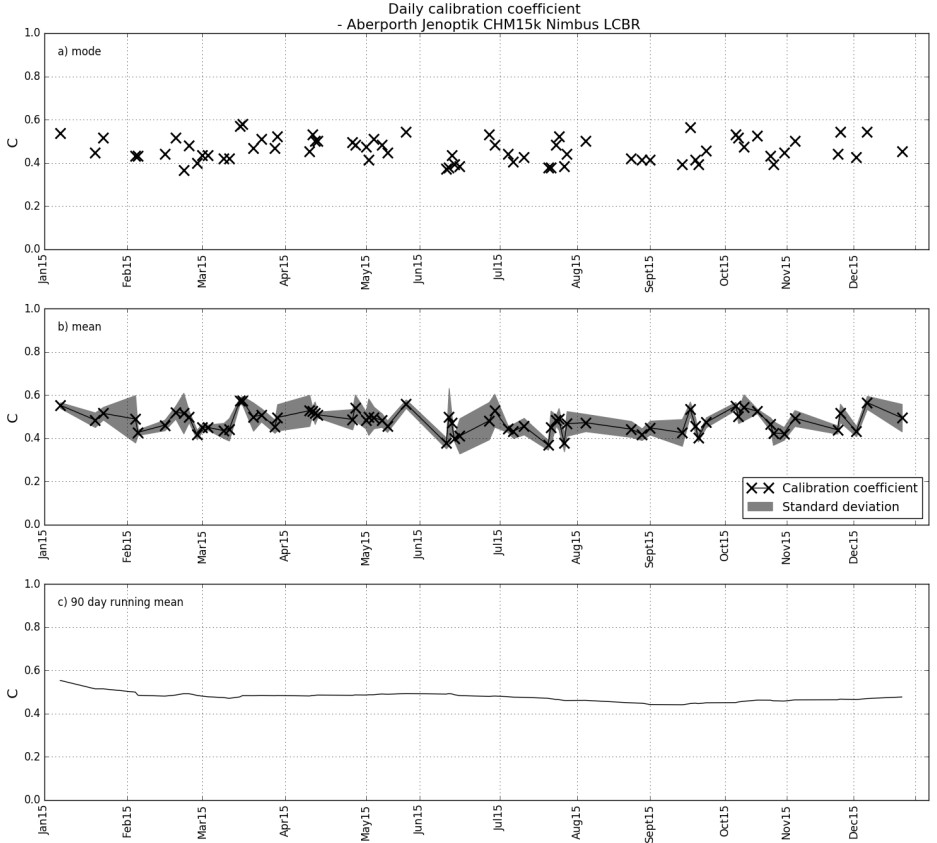

Figure 12: Calibration coefficients for the Lufft CHM15k nimbus ceilometer at Aberporth (52.06° N, 4.33° W).
Each black cross represents a single day, calculated from profiles deemed suitable by the calibration algorithm.
Panel a) shows the mode of C for each individual day, b) shows the mean of C for each day, with the standard
deviation shaded in grey and c) shows a 90-day running mean for the 12-month period. The average of the daily
modes is $0.46 \pm 0.05$, the average of the daily means is $0.48 \pm 0.05$ and the average of the 90-day running means
is $0.48 \pm 0.02$.

615





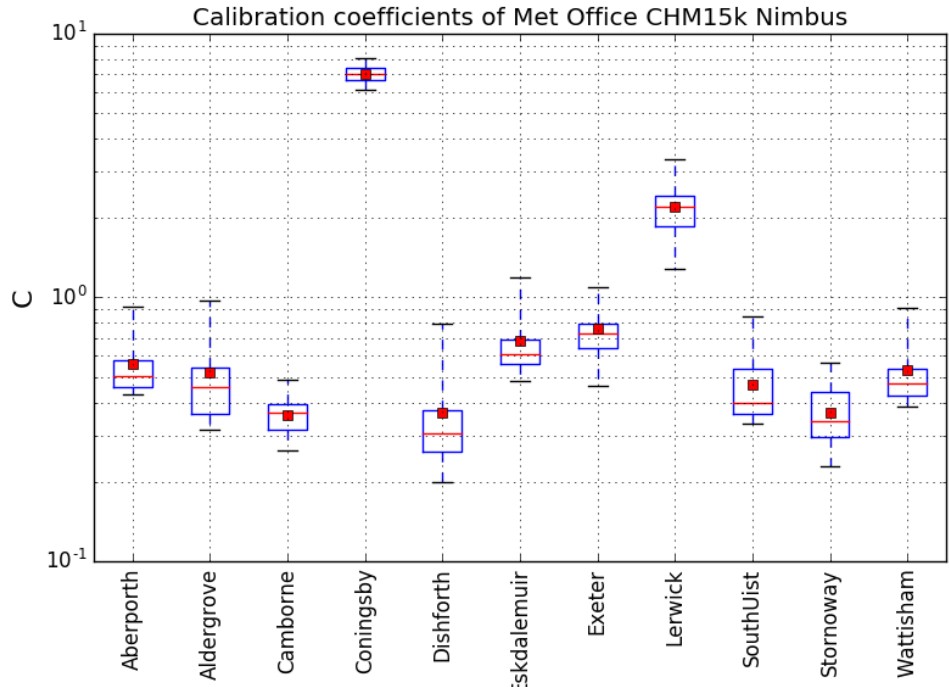

Figure 13: Calibration coefficient for each of the CHM15k Nimbus ceilometers in the UK Met Office network. 3 months of data (January-March 2016) have been used for each instrument. The number of suitable calibration profiles will be dependent on occurrences of cloud and therefore will vary for each instrument. The box outline represents 50 % of the calibration profiles and the whiskers extend to include 95% of the profiles (outliers have been excluded from plot). The horizontal red line in the box shows the median calibration coefficient and the smaller, filled box shows the mean.

620




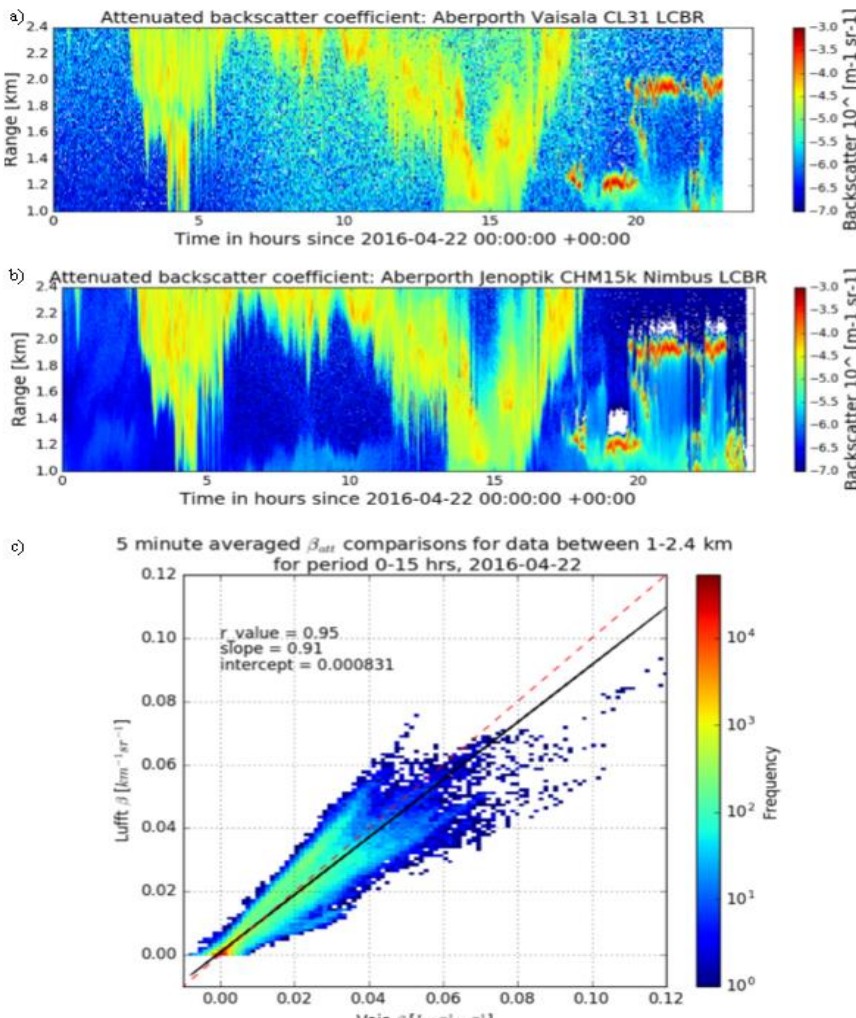

Figure 14: Quicklooks for the observed attenuated backscatter between 1 and 2 km are shown for the (a) Vaisala
ceilometer and the (b) Lufft ceilometer. Panel (c) shows 5-minute averaged attenuated backscatter comparison
for the Lufft and Vaisala ceilometers situated at Aberporth for 22/04/2016 between 0-15 GMT. Colour scale
indicates the number of data points. Vaisala data has been corrected for water vapour attenuation and difference
in wavelength, and has been interpolated to match the resolution of the Lufft ceilometer. The black line shows
the linear fit of the data and the dashed red line is the 1:1 line.