# Peer review of "A robust automated technique for operational calibration of ceilometers using the integrated backscatter from totally attenuating liquid clouds"

_Atmospheric Measurement Techniques, 2018_

## Referee Comment (RC1) · Anonymous Referee #1 · 29 Jan 2019

Hopkin et al. present an automated technique to calibrate ceilometers using liquid clouds.

Ceilometer calibration is not a new topic and this paper is based on already existing methodology (mainly O'Connor et al. 2004). The novel aspect of this work is that the calibration is applied on a long period and for large network with different instrument types. This work is valuable to the scientific community as it can be applied to the new networks that are currently under development.

Therefore I recommend the publication of the manuscript after the following minor revisions.

**1  Specific Comments**

Abstract: the abstract is clearly summarizing the paper. However, I think than one important aspect of the paper is missing. This paper showed that the Lidar signal is affected when the external windows transmission is dropping below 90

Introduction: l 62-67. This paragraph is farfetched. A ceilometer is measuring attenuated backscatter that is not proportional to Ice Water Content. If there is a link between these quantities, please add an equation and the references to support your conclusion. If not, I would suggest I would suggest dropping this 10% requirement in the introduction and the section 5.2.

Section 3 line 204. This is a crucial part of your algorithm. To my understanding, it means that the calibration is valid only when the windows transmission and the energy are higher than 90%. Many instruments will be excluded by this criterion. Please perform a short sensitivity study on this criterion. Please also mention that your calibration is valid only in these conditions and that the authors recommend using the instruments only in this case.

Section 4.1 Wiegner et al (2019) suggested that "error sources beyond the water vapor absorption might be dominant" for CL31. What will be the impact if you drop the water vapor correction? Would you have a better agreement with the Lufft CHM15k?

Section 4.3. line 322. Could you give more details on these outliers? These outliers could be used to identify instrument malfunctions. Section 4.4. could you comment on the high calibration coefficient for Benson and Exeter?

Section 4.4 l355. Again, a quick sensitivity study might be useful to filter precisely the

outliers. Is your calibration still valid when the windows transmission is below 90%? Do you recommend not to use the data when the transmission is low has it affect the signal by 50%. This could be one major conclusion of the paper as it would affect the operational maintenance of ceilometer networks.

Section 5.1: Please add references to the saturation for photon-counting detectors. This is not a new challenge.

Line 381-382: There is a contradiction between this sentence and line 390. Either the negative backscatter can be used to remove the saturated profile, either not. Please replace the sentence "...to remove these profiles" by "...to remove most of these profiles".

Section 5.2 line 416: Do you filter the data at 1km, 2.2km (line 401) or at an height that is instrument specific (line 405)

Section 5.2: l426. Keep the 10% only if the demonstration in the introduction is more robust.

Section 6 line 436. Is the wavelength the only problem? What about the laser power and the instrument sensitivity?

Line 444: Please add a clear reference to support that drizzle scattering is almost wavelength independent

Conclusion: I recommend to add that the calibration is valid only when the windows transmission and the energy pulse are above 90% and that a maintenance should be performed if it is not the case.

**2 Technical corrections**

Introduction: l48. Please mention some operational weather models you are referring to.

Section 2.2 l110 and table 1: please check the maximum range for Vaisala CL31. According to the manual, the typical message contains 770 gates with a 10 meter resolution. Therefore the maximal range is 7.7km.

Section 4. l235. Please mention the Lufft CHM8k that is measuring at 905nm and specify that it the CHM15k that is measuring at 1064nm (l238)

Section 4.2 l279. Could you change the sentence "that confuse the non-expert" by that might confuse the non-expert". One could argue that the cosmetic feature is the confusing process.

L300: please check the reference formatting.

Section 5 line 364: Please mentioned CHM15k. This section is not valid for CHM8k measuring a 905nm.

Section 5.1 l 370 Please mention that the receiver type also has an influence on the saturation ( photon-counting )

---

## Referee Comment (RC2) · Anonymous Referee #2 · 7 Feb 2019

This manuscript presents a methodology for calibrating ceilometers automatically. Although the calibration technique itself is not new, the methodology has been significantly improved so that it accounts for additional sources of error and bias, and demonstrates that it can be applied to more than one type of ceilometer. The methodology has also been extended to be suitable for adoption across a large-scale network, crucial for enabling reliable data production operationally, and also shows the typical performance across a network.

I recommend that this manuscript is suitable for publication after some minor revisions

to clarify some small details.

Specific Comments

Abstract, lines 16-19: Vaisala ceilometers in operational use are often set so that not all signal above 2.4 km has been range-corrected; should you add here a maximum cloud altitude of 2.4 km?

Abstract, line 28: It's not clear which instrument you are referring to, and whether you mean a minimum threshold altitude of 2 km.

Page 2, line 46: It is possible to operate these instruments with 5 m resolution..

Page 3, lines 105-107. This statement is not quite true. It is often preferable to operate at as high a pulse repetition rate as possible in order to increase sensitivity; the limitation is also determined by expected sensitivity in terms of avoiding 'second-trip' echos. For low-powered ceilometer systems, you may not expect to obtain any signals above 15 km even after averaging, hence a high repetition rate of 10 kHz is suitable, whereas high-powered lidar systems are capable of detecting polar stratospheric clouds at 15-25 km, or designed for observing other atmospheric parameters in the stratosphere-mesosphere, and require a lower repetition rate. It may also depend on the detector.

Page 3, line 111, and Table 1: Maximum range of CL31 in standard mode is 7.7 km (770 gates with 10 m vertical resolution, 385 gates with 20 m vertical resolution). If operated at 5 m vertical resolution then maximum range is 7.5 km (1500 gates).

Page 4, line 116: Not airline pilots, but aviation forecasters and air traffic control!

Page 4, lines 136-137: The O'Connor et al. (2004) paper shows that S is constant for droplet size distributions with median volume diameters in the range of 10 to 50 um and states that it starts to fall for distributions with median volume diameters above 50 um.

Page 8, lines 260-261: Do you mean ' at a wavelength of about 910 nm '?

Page 8, line 270 & 273: Suggest that you include NWP or operational forecast, i.e. 'a convection-permitting variable resolution regional NWP model' and 'ECMWF operational forecast model'.

Page 8, lines 276-279: I suggest you expand this to explain that some signals above 2.4 km may not have been range-corrected, which is why they are not suitable.

Page 9, lines 292-302: This feature could be due to the range-dependent multiple scattering factor not being calculated correctly at close ranges (the telescope alignment not being perfect for example).

Page 10, line 340: Could add here that the method works for both coastal locations and inland.

Page 12, line 410: Unlike the Vaisala ceilometers operated in 'standard mode'. The instrument settings can be altered so that this change in processing at 2.4 km is not switched on.

Page 13, line 444: As you state later in the next sentence and the next paragraph, drizzle and and ice scattering aren't wavelength-independent, so I suggest revising this sentence.

Page 13, lines 455-457: Have you checked the impact of multiple scattering for drizzle in your comparison of the two instruments? This may explain why the best fit does not lie on the 1-to-1 line; more multiple scattering would be expected for the Vaisala instrument.

---

## Author Comment (AC1) · 1 Apr 2019

We thank the reviewer for their constructive comments. Our responses are given in the supplement attached below. Responses to the comments are in red and changes to the paper have been made in bold.

Please also note the supplement to this comment:
https://www.atmos-meas-tech-discuss.net/amt-2018-427/amt-2018-427-AC1-supplement.pdf

---

## Author Comment (AC2) · 1 Apr 2019

**Referees' Comments to the Authors**

**Reviewer 1**

We thank Reviewer 1 for his/her constructive comments
Our responses and changes are given below in red; changes to the text are shown in bold

1 Specific Comments

Abstract: the abstract is clearly summarizing the paper. However, I think than one important aspect of the paper is missing. This paper showed that the Lidar signal is affected when the external windows transmission is dropping below 90

Line 20 changed from lidar pulse energy is low to lidar pulse energy falls below 90%

Introduction: l 62-67. This paragraph is farfetched. A ceilometer is measuring attenuated backscatter that is not proportional to Ice Water Content. If there is a link between these quantities, please add an equation and the references to support your conclusion. If not, I would suggest I would suggest dropping this 10% requirement in the introduction and the section 5.2.

New text (line 64) with references added for clarification: **for example, Illingworth et al, (2019) show that a calibration accuracy of 10% is needed when deriving 'O-B' statistics obtained by comparing the observed ceilometer backscatter ('O') from Saharan dust with the forward modelled backscatter ('B') from the ECMWF CAMS model.**

(line 69) **In most models the ice particle density is usually assumed to be inversely proportional to particles size (e.g. Brown and Francis, 1995) so IWC is proportional to extinction and for a given lidar ratio and small amounts of attenuation, the requirement is for a ceilometer calibration accurate to be 10%.**

The attached figure from Illingworth et al (2019) shows the O-B statistics for the ceilometer backscatter observed during a Saharan dust outbreak, and the forward modelled backscatter from the ECMWF CAMS model. The lower right hand panel shows that they agree to within 10% - this conclusion would not have been possible if the ceilometer were not calibrated to within 10%.

[Figure]

Section 3 line 204. This is a crucial part of your algorithm. To my understanding, it means that the calibration is valid only when the windows transmission and the energy are higher than 90%. Many instruments will be excluded by this criterion. Please perform a short sensitivity study on this criterion. Please also mention that your calibration is valid only in these conditions and that the authors recommend using the instruments only in this case.

We have made some further investigations and find that when the window transmission falls then the backscatter trace becomes very noisy. We believe this is because any dust layer on the window is inhomogeneous and whereas the whole window is used for monitoring the window transmission, the actual backscatter signal is only affected by the dust on the central spot of the window that is illuminated by the laser. Accordingly the problem is more serious than implied by the reviewer. When the window is dusty not only is the calibration unreliable but so is the backscatter signal.

We have made the following changes:

1. An extra phrase in the abstract after line 33 stating: **In all cases, if quantitative and reliable backscatter observations are to be obtained it is essential to keep the window clean. This may be a challenge in dusty locations.**

2. Extra sentence after line 209 at the end of the paragraph c:

**For quantitative calibrations and observations of backscatter it is essential that the window be kept clean. It may be possible to correct the observed backscatter for low pulse energy, but seems most unlikely that corrections can be made for the low window transmission because any dust or dirt covering on the window is probably not homogenous. It may be difficult to keep the window clean in locations where dust is common.**

3. In the conclusions we have added after line 503:

**Instrument malfunction can be identified by sudden changes in calibration. If either the window transmission or the pulse energy falls below 90% of normal values, then clearly the instrument sensitivity and the calibration will be different. When the window transmission falls below 90% the backscatter becomes noisy, probably due to the inhomogeneous nature of the layer of dust or dirt on the window, consequently it is essential that the window is kept clean if reliable data are to be obtained. If the pulse energy falls below 90% then it should be possible to correct the backscatter signal but the precise accuracy of this technique remains to be determined.**

Section 4.1 Wiegner et al (2019) suggested that "error sources beyond the water vapor absorption might be dominant" for CL31. What will be the impact if you drop the water vapor correction? Would you have a better agreement with the Lufft CHM15k?

Without the water vapour correction, there is an annual cycle of change in calibration of about 12 %. (see Section 4.1, third paragraph: **for a water vapour path of 2 cm, a typical summer value in the UK, the change in water vapour transmission varies from about 0.77 to 0.75 (about 3%) as the peak laser emissivity increases from 900 to 916 nm. The typical water vapour path in winter is 1cm leading to a transmission of about 0.85, so if no water vapour correction was made, one would expect an apparent annual cycle of the calibration coefficient of about 12%.** ) To highlight this

annual cycle which would be introduced to the calibration if the water vapour absorption was not corrected for, we have also added this point to the caption of figure 7.

Section 4.3. line 322. Could you give more details on these outliers? These outliers could be used to identify instrument malfunctions.

New text added line 339: **The 2% of outliers all have high lidar ratios; they are probably from occasional profiles that do not completely attenuate the lidar signal.**

Section 4.4. could you comment on the high calibration coefficient for Benson and Exeter?

New text added, line 352: **The anomalously high calibration coefficients for Benson and Exeter are probably due to some unknown instrument malfunction as the window transmission and transmit power are recorded as normal. The large value of the calibration coefficient is correcting for this effect, but also flags that there is a malfunction in the instrument.**

Section 4.4 l355. Again, a quick sensitivity study might be useful to filter precisely the C2 outliers. Is your calibration still valid when the windows transmission is below 90%? Do you recommend not to use the data when the transmission is low has it affect the signal by 50%. This could be one major conclusion of the paper as it would affect the operational maintenance of ceilometer networks.

See response above for comment on Section 3 line 204

Section 5.1: Please add references to the saturation for photon-counting detectors. This is not a new challenge.

We have looked for an appropriate reference but apart from our personal communication, we cannot find one. Do you have a recommended reference we could use?

Line 381-382: There is a contradiction between this sentence and line 390. Either the negative backscatter can be used to remove the saturated profile, either not. Please replace the sentence ". . .to remove these profiles" by ". . .to remove most of these profiles".

Contradiction removed and reworded for clarification (line 409): **To increase confidence that only unsaturated profiles are used, a cloud height threshold was also imposed, as demonstrated in Figure 11; this shows a histogram of the uncalibrated integrated attenuated backscatter for profiles in liquid water cloud at Aberporth.**

Section 5.2 line 416: Do you filter the data at 1km, 2.2km (line 401) or at an height that is instrument specific (line 405)

This has been clarified by the addition of the sentence (line 437) **The lower height limit is often higher than 1 km, however, due to the instrument-specific region of saturation**.

(Clouds lower than 1km should not be used for the calibration of the Met Office Lufft CHM15kdue to overlap but these profiles would all be filtered out anyway due to removing profiles that cause saturation).

Section 5.2: l426. Keep the 10% only if the demonstration in the introduction is more robust.

See response and discussion on page one and the figure

Section 6 line 436. Is the wavelength the only problem? What about the laser power and the instrument sensitivity?

Additional text added (line 455): **different power, and different detector sensitivities**

Line 444: Please add a clear reference to support that drizzle scattering is almost wavelength independent

Two references added (line 465) **: Drizzle drops are large compared to the wavelength of the lidar and hence the scattering is almost wavelength-independent for 1064 and 910nm lidars (since we are close to the geometric optics regime), as shown by Westbrook et al. (2010b).**

**The code of Hogan (2006) confirms that for drizzle, multiple scattering is negligible.**

Conclusion: I recommend to add that the calibration is valid only when the windows transmission and the energy pulse are above 90% and that a maintenance should be performed if it is not the case.

Added (see the discussion on page 2 of this reply)

2 Technical corrections

Introduction: l48. Please mention some operational weather models you are referring to.

Some examples added: **Operational weather forecasting models such as those operated by the ECMWF and MeteoFrance**

Section 2.2 l110 and table 1: please check the maximum range for Vaisala CL31. According to the manual, the typical message contains 770 gates with a 10 meter resolution. Therefore the maximal range is 7.7km.

Corrected to 7.7 km

Section 4. l235. Please mention the Lufft CHM8k that is measuring at 905nm and specify that it the CHM15k that is measuring at 1064nm (l238)

Changed to specify that the Lufft CHM15k operate at 1064 nm so do not need correcting for water vapour absorption but that the Lufft CHM8k operate at 905 nm and so do require a correction.

Section 4.2 l279. Could you change the sentence "that confuse the non-expert" by that might confuse the non-expert". One could argue that the cosmetic feature is the confusing process.

Line 293  Changed as recommended to **that might confuse.**

L300: please check the reference formatting.

Corrected

Section 5 line 364: Please mentioned CHM15k. This section is not valid for CHM8k measuring a 905nm.

Added CHM15k

Section 5.1 l 370 Please mention that the receiver type also has an influence on the saturation ( photon-counting )

Type of receiver added in

**Reviewer 2**

We thank Reviewer 2 for his/her constructive comments
Our responses and changes are given below in red; changes to the text are shown in bold

Specific Comments

Abstract, lines 16-19: Vaisala ceilometers in operational use are often set so that not all signal above 2.4 km has been range-corrected; should you add here a maximum cloud altitude of 2.4 km?

We have added '**care must be taken to only use profiles where the range correction is implemented'**.

Abstract, line 28: It's not clear which instrument you are referring to, and whether you mean a minimum threshold altitude of 2 km.

We have rewritten this section so that it reads '**The more sensitive ceilometer model operating at 1064 nm is unaffected by water vapour attenuation but is more prone to saturation in liquid clouds; such profiles can be recognised and rejected and, despite  the more restricted sample of cloud profiles, a robust calibration is readily achieved. In the UK, the running mean 90-day calibration coefficients varied by about 4% over a period of one year.'**

Page 2, line 46: It is possible to operate these instruments with 5 m resolution.

This has been changed to **5 m** resolution

Page 3, lines 105-107. This statement is not quite true. It is often preferable to operate at as high a pulse repetition rate as possible in order to increase sensitivity; the limitation is also determined by expected sensitivity in terms of avoiding 'second-trip' echoes. For low-powered ceilometer systems, you may not expect to obtain any signals above 15 km even after averaging, hence a high repetition rate of 10 kHz is suitable, whereas high-powered lidar systems are capable of detecting polar stratospheric clouds at 15-25 km, or designed for observing other atmospheric parameters in the stratosphere-mesosphere, and require a lower repetition rate. It may also depend on the detector.

We have removed the line '**Due to the low power of ceilometers, they have a much higher pulse repetition rate compared to high-power lidars, to compensate for this lower power and to increase the signal to noise ratio.**' And replaced it with the line '**Ceilometers generally operate at low power, so, to increase signal-to-noise ratio, they tend to have higher pulse repetition rates compared to high-power lidars; the returns from distant signals are generally very low so 'second trip' echoes are not usually a problem.'**

Page 3, line 111, and Table 1: Maximum range of CL31 in standard mode is 7.7 km (770 gates with 10 m vertical resolution, 385 gates with 20 m vertical resolution). If operated at 5 m vertical resolution then maximum range is 7.5 km (1500 gates).

The table has been corrected to read **7.7 km**. We have also clarified in the legend that The Met Office CL31 ceilometers have a vertical resolution of 20 m, apart from at Exeter CL31 where the vertical resolution is 10 m.

Page 4, line 116: Not airline pilots, but aviation forecasters and air traffic control!

Airline pilots corrected to **aviation forecasters and air traffic contro**l

Page 4, lines 136-137: The O'Connor et al. (2004) paper shows that S is constant for droplet size distributions with median volume diameters in the range of 10 to 50 um and states that it starts to fall for distributions with median volume diameters above 50 um.

Corrected from 100 um to 50 um

Page 8, lines 260-261: Do you mean ' at a wavelength of about 910 nm '?

Deleted 'between' from sentence

Page 8, line 270 & 273: Suggest that you include NWP or operational forecast, i.e. 'a convection-permitting variable resolution regional NWP model' and 'ECMWF operational forecast model'.

Changed to 'a convection-permitting variable resolution regional NWP model' and 'ECMWF operational forecast model'.

Page 8, lines 276-279: I suggest you expand this to explain that some signals above 2.4 km may not have been range-corrected, which is why they are not suitable.

Sentence expanded to read **'This is done to avoid the background noise signal leading to apparent clouds at high altitudes that confuse the non-expert, so for the calibration procedure, profiles above 2.4 km are not suitable as the return signal may not have been range corrected'**

Page 9, lines 292-302: This feature could be due to the range-dependent multiple scattering factor not being calculated correctly at close ranges (the telescope alignment not being perfect for example).

Inserted line **311 'We suspect that this is a result of the instrument detector saturating or range dependent multiple scattering not being calculated correctly at close ranges because of imperfect telescope alignment.'**

Page 10, line 340: Could add here that the method works for both coastal locations and inland.

Added in '**from both coast and inland sites'** at end of paragraph.

Page 12, line 410: Unlike the Vaisala ceilometers operated in 'standard mode'. The instrument settings can be altered so that this change in processing at 2.4 km is not switched on.

Reference to Vaisala ceilometers removed from this line to avoid confusion. Line 430 change to **The Lufft ceilometers have a range correction applied to the full attenuated backscatter profile, therefore they are not restricted by a change in processing at 2.4 km**

Page 13, line 444: As you state later in the next sentence and the next paragraph, drizzle and ice scattering aren't wavelength-independent, so I suggest revising this sentence.

Line 465 Reworded to remove contradiction and two references added**: Drizzle drops are large compared to the wavelength of the lidar and hence the scattering is almost wavelength-independent for 1064 and 910nm lidars (since we are close to the geometric optics regime), as shown by Westbrook et al. (2010b).**

Page 13, lines 455-457: Have you checked the impact of multiple scattering for drizzle in your comparison of the two instruments? This may explain why the best fit does not lie on the 1-to-1 line; more multiple scattering would be expected for the Vaisala instrument.

Impact of multiple scattering in drizzle referenced – '**The code of Hogan (2006) confirms that for drizzle, multiple scattering is negligible.'**